# The effect of COVID-19 vaccine tele-educational program on vaccine hesitancy and receiving the vaccine among women planning for pregnancy, pregnant or breast-feeding mothers

Aaliyah Momani[1], Shaher H. Hamaideh[2], Arwa B. Masadeh[1], Fadwa Alhalaiqa[3], Fadi N. Bani Mostafa[4], Haneen Isam Weld Ali[5], Rami Masa'Deh[1]*

1 School of Nursing, Applied Science Private University, Amman, Jordan, 2 Community and Mental Health Nursing Department, Faculty of Nursing, The Hashemite University, Zarqa, Jordan, 3 Faculty of Nursing, Philadelphia University, Philadelphia, Jordan, 4 KLD Management Training LLC, Riyadh, Saudi Arabia, 5 Ministry of Education, Amman, Jordan

* r_masadeh@asu.edu.jo

## Abstract

### Background

COVID-19 hesitancy among women planning to become pregnant, who are pregnant, and who are breast-feeding is still a global phenomenon. Unfortunately, there is a lack of national educational programs that provide those groups of people with the information they need about the vaccine.

### Objective

This study investigated the effect of the COVID-19 vaccine tele-educational program on vaccine hesitancy and receiving the vaccine among women planning for pregnancy, pregnant and breast-feeding mothers.

### Methods

This study implemented a quasi-experimental pre-post design and was conducted in Jordan. It was a two-time study and had two groups of women; 220 women participated in the control group, and 205 women participated in the intervention group (those received the tele-educational program). All participating women answered the demographic characteristics sheet and the Arabic version of Hesitancy About COVID-19 Vaccination Questionnaire twice.

### Results

Results showed that after conducting the program the interventional group reported significantly higher vaccination rate and lower mean score of hesitancy than the control group (M = 24.67, SD = 5.11; M = 27.45, SD = 4.92; respectively) t (423) = -4.116, p-value < 0.001.

**Data Availability Statement:** All relevant data are within the paper and its Supporting Information files.

**Funding:** This research project was financially funded by the Applied Science Private University, Amman, Jordan. The funders had no role in study design, data collection and analysis, decision to publish, or preparation of the manuscript.

**Competing interests:** The authors have declared that no competing interests exist.

Moreover, before the program, women in the intervention group reported significantly higher levels of hesitancy compared to those in the same group after the program (M = 28.35, SD = 4.91; M = 24.66, SD = 5.11; respectively) t (204) = 17.83, p-value < 0.001.

## Conclusions

The study concluded that awareness of pregnant women after being given the tele-education program about COVID-19 vaccination decreased their hesitancy and improved their willingness to participate in the COVID-19 vaccination. Therefore, health workers should focus on providing scientific-based information about the vaccine to reduce the doubts of pregnant women about participating in the COVID-19.

## 1. Introduction

Coronavirus disease 2019 (COVID-19) https://en.wikipedia.org/wiki/Contagious_disease has spread worldwide, become pandemic and affected people's lives in several ways [1–3]. Several measures have been put in place to manage the pandemic. This included lockdowns, social distancing measures, travel restrictions, and vaccine development. Vaccines against infectious disease is considered a first line of prevention [4]. One study in Jordan estimated that 36.4% of 2,208 participants were not willing to take COVID-19 vaccine [5]. Since the revolution of vaccines against infectious diseases, the phenomenon of vaccine hesitancy has been reported around the globe [6–8], and this was not an exception for COVID-19 vaccination [9, 10]. A narrative review including 114 countries/territories that focused on COVID-19 vaccine acceptance rates globally showed that vaccine hesitancy is a global issue. The review reported that in Jordan there was a decline in the rate of COVID-19 vaccination over time [11]. The World Health Organization (WHO) considered vaccine hesitancy one of the top ten global health threats [12]. Of 1,607 participants in one study in Jordan, 55.5% of them had an intention to take COVID-19 vaccine [13]. It was reported that COVID-19 vaccine hesitancy was common in the general population and even more common among specific groups of people such as parents to vaccinate their children, pregnant and breast-feeding women [14, 15]. One study in Jordan found that vaccine hesitancy was higher among non-medical individuals compared to individuals in the medical field [16].

Global studies found that pregnant women; women who were planning to become pregnant; or breast-feeding mothers were more hesitant to take COVID-19 vaccine compared to other women [17–21]. For example, a large-scale study of 145,424 pregnant women from Scotland showed that COVID-19 vaccine coverage was lower in pregnant women (32.3%) than in the general female population (77.4%) [17]. Similarly, in Saudi Arabia, a study of 431 women showed that pregnant women or who were planning to become pregnant were significantly more hesitant, had lower adherence levels, and received fewer COVID-19 vaccines than other women [18]. Three main reasons could explain COVID-19 vaccine hesitancy among this specific population. Firstly, there were concerns regarding COVID-19 vaccine safety, effects, and side effects, mainly because it was developed in a short period of time and received accelerated approval for use [18]. This created concerns about causing harm to the fetus for pregnant women or infertility for women who were planning to become pregnant [14]. Secondly, lack of proper scientific-based knowledge on the seriousness of COVID-19 disease, its effects on pregnancy and breast-feeding, and how the COVID-19 vaccine can minimize the severe effects of the disease on pregnant and breast-feeding women [22, 23]. Thirdly, the rumors and

misinformation announced on the media, the conspiracy theory, and the widespread social uncertainty regarding the vaccine brought a public challenge to receive the vaccine [24].

Based on risk-benefit calculations, it is recommended that pregnant women receive the COVID-19 vaccine [25]. Even though COVID-19 vaccination is recommended for women who are planning to become pregnant, women who are pregnant, and breast-feeding mothers, vaccine hesitancy is still a global phenomenon [4, 26]. In order to overcome the previously mentioned causes of hesitancy, there is a universal need to educate people about the seriousness of COVID-19 disease, the safety and effectiveness of available COVID-19 vaccines, and provide them with the trusted sources of information about the COVID-19 vaccine, particularly for pregnant women, those who are planning for pregnancy or breast-feeding mothers [26].

Recently, there has been a massive international effort to increase public awareness by educating people about COVID-19 vaccination, aiming to maximize the uptake of the vaccine and minimize the barriers to receiving it [27, 28]. In the same line, some authors focused on attending to negative emotions such as fear and anxiety, raising awareness of emotional manipulations by 'anti-vaccine' disinformation efforts, and activating positive emotions such as hope as part of vaccine education endeavors [29]. In Egypt, the effect of the COVID-19 educational program on pregnant women's knowledge, attitude, and practices toward COVID-19 was examined. In one study, authors used telehealth and WhatsApp nursing educational programs using messages and then assessed pregnant women's knowledge, attitude, and practices toward COVID-19 [30]. Studies found that COVID-19 knowledge, attitude, and practices significantly improved after implementing the educational program [30, 31]. Even though there is an international effort to increase awareness about the vaccine, there are scarce resources in the literature on providing an educational program to improve hesitancy toward COVID-19 vaccine among pregnant, planning for pregnancy, or breast-feeding mothers.

To the researchers' knowledge, no interventional educational study was conducted in Jordan that focused on vaccine hesitancy and receiving the vaccine among pregnant, planning for pregnancy, or breast-feeding mothers. Therefore, the main aim of the study was to investigate the effect of a tele-educational program on vaccine hesitancy and receiving the vaccine among women planning for pregnancy, pregnant and breast-feeding mothers. The study was intended to answer the following research question:

- What is the effect of a tele-educational program on vaccine hesitancy and receiving the vaccine among women planning for pregnancy, pregnant and breast-feeding mothers in Jordan?

## 2. Methods

### 2.1. Study design

This study implements a quasi-experimental pre-post design.

### 2.2. Setting

Jordan has twelve governorates; the study was conducted in the three major governorates in Jordan (Amman, Irbid, and Zarqa). These three governorates include more than three-quarters of the Jordanian population [32]. In addition, several prenatal/maternal clinics representing governmental, private, and educational health sectors were involved in this study.

### 2.3. Population, sampling strategy, and sample size

This study had two groups of women; the control group (those who did not attend the intervention) and the intervention group (those who attended the tele-education program). The

target population was all adult women planning for pregnancy, pregnant, or breast-feeding mothers in Jordan. A non-probability convenience sampling technique was used to recruit women. The inclusion criteria were women above 18 years old, who did not receive the COVID-19 vaccine, agreed to participate in the two phases of the study and were able to read, write and understand Arabic.

The sample size was calculated using G power software [33]. Considering the tests used in this study, a power of 0.80, 0.05 level of significance, and a medium effect size of 0.5, 128 women in each group were enough to get statistically significant results. However, the researcher found many women who met the inclusion criteria and expressed their interest in participating in the study. By the end of the data collection period, between December 2021 and April 2022, 220 women participated in the control group, and 205 women participated in the intervention group.

## 2.4. Measurements of variables

**2.4.1. The demographic characteristics sheet.** It includes information regarding the age of women; residency, educational status; marital status; history of chronic diseases; previous infection with COVID-19; and financial status.

**2.4.2. The Arabic version of hesitancy about COVID-19 vaccination questionnaire.** This questionnaire has been previously used in studies around the globe and is reported to be a valid and reliable questionnaire to be used in Arabic communities [18, 34]. In this questionnaire, hesitancy receiving the vaccine includes nine items. Each item was scored using a five-point Likert scale ranging from one (strongly disagree) to five (strongly agree). The possible scores ranged from 9 to 45. The higher the score of hesitancy, the greater the hesitancy to receive the vaccine [18, 34].

**2.4.3. Educational program.** Information provided in this program was initially planned to target women planning for pregnancy, pregnant women, and finally, women who breast-fed their children and, more specifically, those who did not take the COVID-19 vaccine. The program includes an individual tele-education (interactive education phone sessions, phone calls consultancy, text message, and digital education booklet) provided to the women in the intervention group for two weeks. Information provided in the educational program was adopted from the CDC [4]. The researcher called participating women in the intervention group and discussed COVID-19 disease, the benefits of the vaccine, effectiveness, side effects of the available vaccines, sources of their information about the vaccine, and answered their questions about the disease and the vaccine. Furthermore, women had the chance to contact the researcher asking for information/further explanation over the phone. The program was administered to women in the intervention group, and no intervention was administered to those in the control group.

## 2.5. Ethical considerations

Ethical approval was obtained from the Institutional Review Board (IRB) at the Applied Science Private University and all organizations from which participants were recruited (IRB number: 2021-2022-1-3). These organizations are prenatal/maternal clinics representing governmental, private, and educational health sectors. Before collecting data, women were informed that participation was voluntary and had the right to withdraw without reason. Also, participants were informed about the anonymity and confidentiality measures taken throughout the study and were assured that all data would be used only for research purposes. Furthermore, no one other than the researcher would have access to their data. Each participant was assigned a numeric code to protect their identity, and coding was used so that questionnaires

of the first and second phases could be linked to each woman. Moreover, a written consent form included the title, purpose, and complete information of the study that women had to sign before being involved in the study. By signing the consent form, women declared that everything was clear to them and willingly agreed to participate in the two phases of the study. Finally, the questionnaires were stored on a secured desktop of the researcher and were only accessible to the principal researcher. Participants did not receive any kind of compensation to participate in this study.

## 2.6. Data collection procedure

After gaining the required ethical approvals, explanatory posters were announced at prenatal/ maternity clinics. The researcher visited these clinics and explained the purpose and the national importance of the study to the women. Those who met the inclusion criteria and were willing to participate signed consent forms, filled in the questionnaires in the study's first phase, and returned them to the researcher. At this point, it was explained that this is a two-phase study, and women will be asked to answer the questionnaire for a second time. Therefore, all participating women had to write their contact details to be called later in the second phase. Furthermore, at this stage, the researcher explained that there was a tele-educational program provided by the researcher as a part of the study, and those who agreed to participate in the program had to tick on the agreement option. By ticking this option, women were included in the education program and were considered in the intervention group. Those who did not choose this option did not attend the program and were considered a control group.

## 2.7. Statistical analysis

Data were analyzed using Statistical Package for Social Science (SPSS) software, version 27 [35]. Categorical variables were reported as frequencies and percentages. Continuous data were reported as mean ± standard deviation (SD). Independent samples t-test was used to compare the mean scores of the hesitancy to take the COVID-19 vaccine between the intervention and the control group in the first phase of the study. This step was conducted to ensure no difference in mean scores of hesitancy taking the COVID-19 vaccine between participants in both groups. Next, a paired-samples t-test was used to compare the mean scores of the hesitancy to take the COVID-19 vaccine in the intervention and control groups pre and post the program. Concerning COVID-19 vaccination, related samples McNemar test was conducted to examine any vaccine rate difference before and after the program. Finally, Wilcoxon signed-rank test was used to investigate any difference in taking the vaccine in the intervention group pre and post the program. A *P*-value of <0.05 was applied to represent the statistical significance of the results, as the level of significance was predetermined as 0.05.

## 3. Results

In this study, 436 women filled in the questionnaires. However, there were five cases were women refused to participate without giving a reason and six women had missing data including the variable of interest and therefore were excluded from the study. At the end, this study included 425 women from different cities in Jordan, with most having a bachelor's degree. Of those, 220 joined the control group, and 205 were in the intervention group. As presented in Table 1, more than two-quarters of them were pregnant or breast feeding their child, and the rest were planning for pregnancy. The majority of the participants had previous infections with COVID-19. Almost 66% of the women have no previous medical history, while some of them, 12.5% had diabetes, and 12.2% had hypertension.

**Table 1. Sociodemographic and clinical characteristics of the women.**

| Variable | Frequency (%) or Mean ± SD | Frequency (%) or Mean ± SD |
|---|---|---|
| | **Intervention** | **Control** |
| **Group:** | 205 (48.24%) | 220 (51.76%) |
| **Are you:** | | |
| Planning to be pregnant | 76 (37.1%) | 80 (36.4%) |
| Pregnant women | 96 (46.8%) | 94 (42.7%) |
| Breast-feeding women | 33 (16.1%) | 46 (20.9%) |
| **Previous COVID-19 infection:** | | |
| Yes | 180 (87.8%) | 194 (88.2%) |
| No | 25 (12.2%) | 26 (11.8%) |
| **Chronic diseases:** | | |
| No medical history | 144 (70.2%) | 140 (63.6%) |
| Hypertension | 28 (13.7%) | 34 (15.5%) |
| Diabetes mellitus | 16 (7.8%) | 27 (12.3%) |
| Immune system disease | 12 (5.9%) | 14 (6.4%) |
| Others | 5 (2.4%) | 5 (2.2%) |
| **Employment:** | | |
| Employed | 105 (51.2%) | 104 (47.3%) |
| Unemployed | 100 (48.8%) | 116 (52.7%) |
| **Residency:** | | |
| Amman | 115 (56.1%) | 159 (72.3%) |
| Zarqa | 47 (36.1%) | 55 (25%) |
| Irbid | 16 (7.8%) | 6 (2.7%) |
| **Level of education:** | | |
| Primary education | 0 (0%) | 1 (0.5%) |
| Secondary education | 11 (5.4%) | 17 (7.7%) |
| Bachelor | 194 (94.5%) | 202 (91.8%) |
| **Age** | 40.73 ± 5.80 | 41.13 ± 5.83 |

## 3.1. The effect of a tele-educational program on vaccine hesitancy and receiving the vaccine among women planning for pregnancy, pregnant and breast-feeding mothers

As presented in Table 2, the intervention group reported significantly lower mean score of hesitancy than the control group after delivering the program (M = 24.67, SD = 5.11; M = 27.45, SD = 4.92. respectively) t (423) = -4.116, p-value < 0.001.

Results showed a statistically significant difference in the levels of hesitancy in the intervention group. Before the program, women in the intervention group reported significantly higher levels of hesitancy compared to those after the program (M = 28.35, SD = 4.91; M = 24.66, SD = 5.11; respectively) t (204) = 17.83, p-value < 0.001. However, results for the control

**Table 2. Hesitancy of getting the vaccine between the intervention group and the control group.**

| Variable | Group | N | Mean ± SD | T | DF | P-value |
|---|---|---|---|---|---|---|
| Hesitancy before the program | Intervention | 205 | 28.35 ± 4.91 | 1.529 | 423 | 0.127 |
| | Control | 220 | 27.61 ± 4.96 | | | |
| Hesitancy after the program | Intervention | 205 | 24.67 ± 5.11 | -4.116 | 423 | <0.001 |
| | Control | 220 | 27.45 ± 4.92 | | | |

**Table 3. Hesitancy of getting the vaccine before and after the program.**

| Variable | | Mean ± SD | Mean difference | T | DF | P-value |
|---|---|---|---|---|---|---|
| Intervention group | Hesitancy before program | 28.35 ± 4.91 | 3.69 | 17.83 | 204 | <0.001 |
| | Hesitancy after program | 24.66 ± 5.11 | | | | |
| Control group | Hesitancy before program | 27.61 ± 4.96 | 0.16 | 1.22 | 219 | 0.22 |
| | Hesitancy after program | 27.45 ± 4.92 | | | | |

group showed no statistically significant difference in the mean score of the hesitancy to get the vaccine before and after the program, see Table 3. This indicates that the program significantly lowered the mean score of hesitating to get the vaccine for the intervention group only.

Results showed that women in the intervention group received the vaccine at a significantly higher rate after the program than before it. However, receiving the vaccine did not differ for the control group, Table 4.

## 4. Discussion

The current study investigated the effect of a tele-educational program on vaccine hesitancy and receiving the vaccine among women planning for pregnancy, pregnant and breast-feeding mothers in Jordan. This study showed that both groups reported similar levels of hesitancy before conducting the program. However, the main result of the current study showed that after delivering the tele-educational program, women in the intervention group had lower rates of COVID-19 vaccine hesitancy and a higher rate of receiving the vaccine than those in the control group.

Consistent with the results of the current study and despite the international recommendations for COVID-19 vaccination for all adults, many studies reported high levels of hesitancy among individuals with low levels of education, among women who are planning to get pregnant, pregnant women, and breast-feeding mothers [4, 36–39]. This hesitancy was explained by a lack of knowledge about COVID-19 disease, vaccine safety, the importance of the COVID-19 vaccine for women who are planning to get pregnant, pregnant women, or breast-feeding mothers, and the misleading information on social media platforms [4, 40–42]. Furthermore, one study found that insufficient prenatal care was one factor that was associated with COVID-19 vaccine hesitancy [43]. This may suggest that health workers have an important role in patients' education on vaccine safety and importance. Accordingly, previous studies suggested that evidence-based education could change women's behavior, which is likely to decrease COVID-19 vaccine hesitancy and, therefore, minimize the effect of the COVID-19 pandemic [44, 45].

The impact of education programs on COVID-19 hesitancy in military-based population and patients diagnosed with inflammatory bowel disease was examined in two studies [46, 47]. The results showed that the educational programs reduced levels of hesitancy in both studies. These studies suggested a need for providing education to groups based on their situation.

**Table 4. Difference in COVID-19 vaccination.**

| Variable | | Not taken | One dose | P-value |
|---|---|---|---|---|
| Intervention group | COVID-19 vaccination before the program | 205 | 0 | <0.001 |
| | COVID-19 vaccination after the program | 43 | 162 | |
| Control group | COVID-19 vaccination before the program | 220 | 0 | 0.125 |
| | COVID-19 vaccination after the program | 216 | 4 | |

Furthermore, the effect of COVID-19 educational programs on COVID-19 vaccination attitudes has been explored [48]. For example, one study of 501 adults showed that after delivering a video tutorial on COVID-19 vaccination, participants' attitudes toward accepting the vaccine improved, significantly improving COVID-19 perceived knowledge and safety during the COVID-19 outbreak [48]. However, none of the participants of these studies were women planning for pregnancy, currently pregnant, or breast-feeding mothers. It is important to mention here that health workers need educational programs helping them dealing with vaccine hesitancy among patients [49], as well as improving health workers communication with pregnant women on the issue of COVID-19 vaccine safety should be considered [39].

The current study implemented a tele-education program that targeted women planning for pregnancy, pregnant and breast-feeding mothers. The effect of educational programs on COVID-19 on women who are planning to get pregnant, are currently pregnant, or are breast-feeding mothers has been reported in the literature [50]. For example, one study conducted on pregnant women's knowledge, attitude, and practices toward COVID-19 in Egypt showed a decrease in levels of hesitancy after delivering a telehealth educational program [50]. In addition, health education on COVID-19 vaccination provided by health workers to pregnant women's improved their knowledge about the COVID-19 vaccine during pregnancy and the mother's willingness to participate in the COVID-19 vaccination [51]. Moreover, a single education lecture decreased pregnant women's hesitancy [52]. These results were in line with results that emerged in the current study.

### 4.1. Study strengths and limitations

Participants were recruited from three major governorates in Jordan. Furthermore, to the best of the researchers' knowledge, it is the first interventional study conducted in Jordan among the studied participants. Moreover, the current study implemented a quasi-experimental design. Therefore, there may be a need to conduct a randomized clinical trial in future studies.

### 4.2. Conclusions and recommendations

Based on the study results, it was concluded that the knowledge of pregnant women after being given a tele education program about COVID-19 vaccination improved their willingness to participate in the COVID-19 vaccination and decreased their hesitancy. Efforts need to be made to increase the knowledge of pregnant women through health education on COVID-19 vaccination by health workers so that the coverage of COVID-19 vaccination in pregnant women can increase, which in turn will affect the morbidity and mortality rates of pregnant women due to COVID-19 infection. Further studies evaluating individual and social benefits of such educational actions or events, including follow-up, are warranted to evaluate the program's cost-effectiveness. Furthermore, future studies are needed to develop other effective instructional methods directed to specific groups of people and, based on their challenges, aim to overcome vaccine hesitancy. Although the end of the COVID-19 pandemic is still unknown, increasing adherence to vaccination should hasten the end of the pandemic. Therefore, health authorities in Jordan (and around the globe) should work in conducting larger follow-up studies not only among pregnant women but also among the general public in the country.

### Supporting information

**S1 File.**
(SAV)

## Acknowledgments

We would like to thank participants for their valuable participation.

## Author Contributions

**Conceptualization:** Fadwa Alhalaiqa.

**Data curation:** Shaher H. Hamaideh, Fadwa Alhalaiqa, Haneen Isam Weld Ali, Rami Masa'Deh.

**Formal analysis:** Shaher H. Hamaideh, Rami Masa'Deh.

**Methodology:** Aaliyah Momani, Arwa B. Masadeh, Fadi N. Bani Mostafa, Haneen Isam Weld Ali, Rami Masa'Deh.

**Supervision:** Haneen Isam Weld Ali.

**Writing – original draft:** Aaliyah Momani, Arwa B. Masadeh, Rami Masa'Deh.

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
