## [Editor Report · Decision Letter 0]

8 Jan 2023

PONE-D-22-31498The effect of COVID-19 vaccine tele-educational program on vaccine hesitancy and receiving the vaccine among women planning for pregnancy, pregnant or breast-feeding mothersPLOS ONE

Dear Dr. Masa'Deh,

Thank you for submitting your manuscript to PLOS ONE. After careful consideration, we feel that it has merit but does not fully meet PLOS ONE’s publication criteria as it currently stands. Therefore, we invite you to submit a revised version of the manuscript that addresses the points raised during the review process. comments and notes: 

1. please adhere to the submission guidelines regarding spacing in the article format

2. Please Include page numbers and line numbers in the manuscript file.

3. please use the correct referencing style as outlined in the submission guidelines

4. please correct the Manuscript Organization style of the manuscript as per the submission guidelines

Manuscripts should be organized as follows. Instructions for each element appear below the list.

Beginning section

The following elements are required, in order:

Title page: List title, authors, and affiliations as first page of the manuscript

Abstract

Introduction

Middle section

The following elements can be renamed as needed and presented in any order:

Materials and Methods

Results

Discussion

Conclusions (optional)

Ending section

The following elements are required, in order:

Acknowledgments

References

Supporting information captions (if applicable)

Other elements

Figure captions are inserted immediately after the first paragraph in which the figure is cited. Figure files are uploaded separately.

Tables are inserted immediately after the first paragraph in which they are cited.

Supporting information files are uploaded separately.

5. correct the following spelling mistakes:

page 5, line 4 vaccination instead of vaccines

page5, line 10 change declining to decline

page6, line 7 correct spelling of COVID-19

page6, line 32 change Statisitics to Statistics

6. regarding introduction: please consider Include a brief review of the key literature and making it more precise and concise, with less wording.

7. avoid overstating the conclusions. Authors may discuss possible implications for their results as long as these are clearly identified as hypotheses instead of conclusions

8. References are listed at the end of the manuscript and numbered in the order that they appear in the text. In the text, cite the reference number in square brackets.

9. make sure to read the submission guidelines in details and correct the outline of the manuscript

10. We recommend that authors seek independent editorial help before submitting a revision. These services can be found on the web using search terms like “scientific editing service” or “manuscript editing service.”

11. 1. please consider adding the following articles of a similar topic and discussed the topic you are discussing and published form Jordan:

A. Aloweidi A, Bsisu I, Suleiman A, et al.: Hesitancy towards COVID-19 vaccines: an analytical cross-sectional study. Int J Environ Res Public Health. 2021, 18:5111. 10.3390/ijerph18105111

B. Abdullah M, Shahait A D, Qaisieh R, et al. (June 26, 2022) Perspectives on COVID-19 Vaccines and Its Hesitancy Among Jordanian Population. Cureus 14(6): e26337. doi:10.7759/cureus.26337

We look forward to receiving your revised manuscript.

Kind regards,

Mohamad Abdullah, M.D., CHM.G/S., MRCS.

Academic Editor

PLOS ONE

Journal Requirements:

3. Thank you for stating the following in the Funding statement Section of your manuscript: 

"This research project was funded by the Applied Science Private University, Amman, Jordan."

"The funder had no role in study design, data collection and analysis, decision to publish, or preparation of the manuscript."
---

## [Author Response · Author response to Decision Letter 0]

22 Jan 2023

Response to reviewers are submitted in a separate file

---

## [Decision Letter · Decision Letter 1]

14 Feb 2023

PONE-D-22-31498R1The Effect of COVID-19 Vaccine Tele-educational Program on Vaccine Hesitancy and Receiving the Vaccine Among Women Planning for Pregnancy, Pregnant or Breast-feeding MothersPLOS ONE

Dear Dr. Masa'Deh,

Thank you for submitting your manuscript to PLOS ONE. After careful consideration, we feel that it has merit but does not fully meet PLOS ONE’s publication criteria as it currently stands. Therefore, we invite you to submit a revised version of the manuscript that addresses the points raised during the review process.

Thank you for submitting this article to PLOS one journal, this article has been reviewed by two reviewers, you can find the reviewer comments bellow.  please read the comments carefully and make the required changes according to the reviewer comments

We look forward to receiving your revised manuscript.

Kind regards,

Mohamad Abdullah, M.D., CHM.G/S., MRCS.

Academic Editor

PLOS ONE

Journal Requirements:

Additional Editor Comments (if provided):

comments from reviewers :

reviewer 1:

The article is presented in a logical and scientific way. Although the language needed to be enhanced to a standard level. Data availability section should be added.

This study has been performed in 2021, I suggest the authors should go and see the state of the art in the topic and enrich the introduction and discussion.

The sample is not quite representative as doesn't include women from south Jourdan.

reviewer 2:

This study has a significant added value to the literature, as it is one of the first interventional studies to investigate the effect of a tele-educational program on vaccine hesitancy among pregnant, planning for pregnancy, or breast-feeding mothers and examined detailed COVID-19 vaccine acceptance among such a vulnerable group in an Eastern Mediterranean developing country. The manuscript is well-written, and the study findings would interest the readers of PlosOne journal. The title and abstract cover the critical aspects of the work. The Introduction provides background and information relevant to the study. The methods are straightforward and replicable. The results are novel and easy to follow, and the statistical tests used are appropriate and answer the study questions. However, there are some comments and suggestions that are doable and will improve the quality of the article if done appropriately.

My main concern is that many of the analysis steps and tests are described in the Results section rather than the Methods section of the manuscript. Also, the results section is not the right place for conclusions. The Results section should include the study findings without details on the study strategy, analysis steps, or conclusions. This point is described in detail in the following comments.

Introduction:

1. The first paragraph of the Introduction is too long. Thus, I encourage the authors to divide the Introduction into 2 or 3 paragraphs rather than one paragraph. For example:

The second paragraph could be started with “Global studies found that pregnant women ………”

The third paragraph from “Based on risk-benefit calculations……….”

2. Also, I prefer to include in the first paragraph of the Introduction section more information about seroprevalence and vaccine coverage worldwide and in developing countries. Also, you could mention the benefits of vaccination over traditional precautionary measures, such as curfews, isolation, sealing the borders …..etc. These benefits could be included in the Introduction and Discussion sections of the manuscript. You could use the following references:

A. https://ourworldindata.org/covid-vaccinations

B. Sheikh Ali S, Kheirallah KA, Sharkas G, et al. SARS-CoV-2 Seroepidemiological Investigation in Jordan: Seroprevalence, Herd Immunity, and Vaccination Coverage. A Population-Based National Study. Int J Gen Med. 2022;15:7053-7062. Published 2022 Sep 5. doi:10.2147/IJGM.S371711.

C. Ramadan M, Hasan Z, Saleh T, et al. Beyond knowledge: Evaluating the practices and precautionary measures towards COVID-19 amongst medical doctors in Jordan. Int J Clin Pract. 2021;75(6):e14122. doi:10.1111/ijcp.14122.

D. Al-Mistarehi AH, Kheirallah KA, Yassin A, et al. Determinants of the willingness of the general population to get vaccinated against COVID-19 in a developing country. Clin Exp Vaccine Res. 2021;10(2):171-182. doi:10.7774/cevr.2021.10.2.171

E. Samrah SM, Al-Mistarehi AW, Ibnian AM, et al. COVID-19 outbreak in Jordan: Epidemiological features, clinical characteristics, and laboratory findings. Ann Med Surg (Lond). 2020;57:103-108. Published 2020 Jul 18. doi:10.1016/j.amsu.2020.07.020

Methods:

3. In the Ethical considerations subsection, could you clarify the organizations from which participants were recruited?

4. Did the participants receive any compensation or rewards for participating in the study? Please, include this point in the manuscript.

5. Are there any missing data during the collection process? If yes, how did you deal with it?

6. Do the authors have data about the number of women who refused to participate and their reasons?

Results:

7. In the first paragraph, you mentioned again, “Before the program, none of them received the COVID-19 vaccine.” This point is redundant for the inclusion criteria, and I prefer to remove it here.

8. Could you calculate and add the p-values to table 1 to compare the two groups? Also, please raise up the age variable in table 1 to the third raw (Directly after the group variable)

9. This paragraph is redundant for what was previously mentioned in the Methods: “Three analysis steps were done to determine the effect of a tele-educational program on vaccine hesitancy among women planning for pregnancy and pregnant and breast-feeding mothers. Firstly, an independent t-test was used to examine any difference in the mean score of the hesitancy to get the vaccine before conducting the program between the intervention group and the control group” >>> The Results section should include the study findings with no details on the study strategy and analysis steps. These points should be just mentioned in the Methods section of the manuscript.

10. Similarly, the following paragraph should be omitted from the Results section: “Secondly, an independent t-test was used to examine any difference in the mean score of the hesitancy to get the vaccine between the intervention group and the control group after conducting the program.”

11. In table 2, the p-value of 000 should be presented as (<0.001). As well as, the p-values in table 3 and table 4 should be corrected.

12. Also, the following paragraph is redundant and should be omitted from the Results section: “Thirdly, a paired samples t-test was conducted to examine any difference in the mean score of the hesitancy to get the vaccine for the intervention group and control group before and after the program.”

13. This paragraph should be transferred from the Results section to the Methods section: “Concerning COVID-19 vaccination, related samples McNemar test was conducted to examine any vaccine rate difference before and after the program.”

14. The last paragraph in the Results section should be transferred to the Discussion and Conclusion sections of the manuscript: “To conclude, there was no significant difference in the hesitancy of COVID-19 vaccination …………………… of COVID-19 vaccination and increasing the tendency to get the vaccine in the interventional group only.”

Discussion:

15. The discussion should be extended with more details to include more information about vaccination hesitancy and determinants in developing countries, particularly among women planning to be pregnant, and how this study will solve this significant issue. Also, as previously mentioned, I prefer to include the benefits of vaccination over traditional precautionary measures, such as curfews, isolation, sealing the borders …..etc. In addition, this study’s findings should be compared with previous studies in the literature.

I would refer you to the following helpful references:

a. Al-Mistarehi AH, Kheirallah KA, Yassin A, et al. Determinants of the willingness of the general population to get vaccinated against COVID-19 in a developing country. Clin Exp Vaccine Res. 2021;10(2):171-182. doi:10.7774/cevr.2021.10.2.171

b. Wainstock T, Sergienko R, Orenshtein S, Sheiner E. Factors associated with COVID-19 vaccination likelihood during pregnancy [published online ahead of print, 2023 Jan 18]. Int J Gynaecol Obstet. 2023;10.1002/ijgo.14680. doi:10.1002/ijgo.14680

c. Blakeway H, Prasad S, Kalafat E, et al. COVID-19 vaccination during pregnancy: coverage and safety. Am J Obstet Gynecol. 2022;226(2):236.e1-236.e14. doi:10.1016/j.ajog.2021.08.007

d. Bechini A, Moscadelli A, Pieralli F, et al. Impact assessment of an education course on vaccinations in a population of pregnant women: a pilot study. J Prev Med Hyg. 2019;60(1):E5-E11. Published 2019 Mar 29. doi:10.15167/2421-4248/jpmh2019.60.1.1093

16. Study strengths:

a) I do not think the study sample size is one of the strengths.

b) One of its strengths is the study’s quasi-experimental interventional design, timeliness, and target population of women who did not take the COVID-19 vaccine.

c) Most vitally, this is one of the first studies that obtained insights into the effect of a tele-educational program on vaccine hesitancy among pregnant, planning for pregnancy, or breast-feeding mothers and examined detailed COVID-19 vaccine acceptance among such a vulnerable group in an Eastern Mediterranean developing country. Thus, this study tried to fill the gap in the literature regarding such issues outside Western countries.

17. Study limitations:

A. The study sample size is relatively small. However, the sample was collected from different areas throughout the country.

B. The selection bias can not be excluded as the study participants were collected from prenatal/maternity clinics. Thus, the results are unlikely to be generalizable beyond those who responded.

C. Also, the point mentioned in the manuscript: “Although the study aimed to include only pregnant women, planning for pregnancy, and breast-feeding mothers, their answers may vary from the general population and people in other situations.”

Reviewers' comments:

Reviewer's Responses to Questions

**Comments to the Author**

1. If the authors have adequately addressed your comments raised in a previous round of review and you feel that this manuscript is now acceptable for publication, you may indicate that here to bypass the “Comments to the Author” section, enter your conflict of interest statement in the “Confidential to Editor” section, and submit your "Accept" recommendation.

Reviewer #1: (No Response)

Reviewer #2: (No Response)

2. Is the manuscript technically sound, and do the data support the conclusions?

Reviewer #1: Yes

Reviewer #2: Yes

3. Has the statistical analysis been performed appropriately and rigorously? 

Reviewer #1: Yes

Reviewer #2: Yes

4. Have the authors made all data underlying the findings in their manuscript fully available?

Reviewer #1: No

Reviewer #2: Yes

5. Is the manuscript presented in an intelligible fashion and written in standard English?

Reviewer #1: No

Reviewer #2: Yes

6. Review Comments to the Author

Reviewer #1: The article is presented in a logical and scientific way. Although the language needed to be enhanced to a standard level. Data availability section should be added. This study has been performed in 2021, I suggest the authors should go and see the state of the art in the topic and enrich the introduction and discussion. The sample is not quite representative as doesn't include women from south Jourdan.

Reviewer #2: This study has a significant added value to the literature, as it is one of the first interventional studies to investigate the effect of a tele-educational program on vaccine hesitancy among pregnant, planning for pregnancy, or breast-feeding mothers and examined detailed COVID-19 vaccine acceptance among such a vulnerable group in an Eastern Mediterranean developing country. The manuscript is well-written, and the study findings would interest the readers of PlosOne journal. The title and abstract cover the critical aspects of the work. The Introduction provides background and information relevant to the study. The methods are straightforward and replicable. The results are novel and easy to follow, and the statistical tests used are appropriate and answer the study questions. However, there are some comments and suggestions that are doable and will improve the quality of the article if done appropriately.

My main concern is that many of the analysis steps and tests are described in the Results section rather than the Methods section of the manuscript. Also, the results section is not the right place for conclusions. The Results section should include the study findings without details on the study strategy, analysis steps, or conclusions. This point is described in detail in the following comments.

Introduction:

1. The first paragraph of the Introduction is too long. Thus, I encourage the authors to divide the Introduction into 2 or 3 paragraphs rather than one paragraph. For example:

The second paragraph could be started with “Global studies found that pregnant women ………”

The third paragraph from “Based on risk-benefit calculations……….”

2. Also, I prefer to include in the first paragraph of the Introduction section more information about seroprevalence and vaccine coverage worldwide and in developing countries. Also, you could mention the benefits of vaccination over traditional precautionary measures, such as curfews, isolation, sealing the borders …..etc. These benefits could be included in the Introduction and Discussion sections of the manuscript. You could use the following references:

A. https://ourworldindata.org/covid-vaccinations

B. Sheikh Ali S, Kheirallah KA, Sharkas G, et al. SARS-CoV-2 Seroepidemiological Investigation in Jordan: Seroprevalence, Herd Immunity, and Vaccination Coverage. A Population-Based National Study. Int J Gen Med. 2022;15:7053-7062. Published 2022 Sep 5. doi:10.2147/IJGM.S371711.

C. Ramadan M, Hasan Z, Saleh T, et al. Beyond knowledge: Evaluating the practices and precautionary measures towards COVID-19 amongst medical doctors in Jordan. Int J Clin Pract. 2021;75(6):e14122. doi:10.1111/ijcp.14122.

D. Al-Mistarehi AH, Kheirallah KA, Yassin A, et al. Determinants of the willingness of the general population to get vaccinated against COVID-19 in a developing country. Clin Exp Vaccine Res. 2021;10(2):171-182. doi:10.7774/cevr.2021.10.2.171

E. Samrah SM, Al-Mistarehi AW, Ibnian AM, et al. COVID-19 outbreak in Jordan: Epidemiological features, clinical characteristics, and laboratory findings. Ann Med Surg (Lond). 2020;57:103-108. Published 2020 Jul 18. doi:10.1016/j.amsu.2020.07.020

Methods:

3. In the Ethical considerations subsection, could you clarify the organizations from which participants were recruited?

4. Did the participants receive any compensation or rewards for participating in the study? Please, include this point in the manuscript.

5. Are there any missing data during the collection process? If yes, how did you deal with it?

6. Do the authors have data about the number of women who refused to participate and their reasons?

Results:

7. In the first paragraph, you mentioned again, “Before the program, none of them received the COVID-19 vaccine.” This point is redundant for the inclusion criteria, and I prefer to remove it here.

8. Could you calculate and add the p-values to table 1 to compare the two groups? Also, please raise up the age variable in table 1 to the third raw (Directly after the group variable)

9. This paragraph is redundant for what was previously mentioned in the Methods: “Three analysis steps were done to determine the effect of a tele-educational program on vaccine hesitancy among women planning for pregnancy and pregnant and breast-feeding mothers. Firstly, an independent t-test was used to examine any difference in the mean score of the hesitancy to get the vaccine before conducting the program between the intervention group and the control group” >>> The Results section should include the study findings with no details on the study strategy and analysis steps. These points should be just mentioned in the Methods section of the manuscript.

10. Similarly, the following paragraph should be omitted from the Results section: “Secondly, an independent t-test was used to examine any difference in the mean score of the hesitancy to get the vaccine between the intervention group and the control group after conducting the program.”

11. In table 2, the p-value of 000 should be presented as (<0.001). As well as, the p-values in table 3 and table 4 should be corrected.

12. Also, the following paragraph is redundant and should be omitted from the Results section: “Thirdly, a paired samples t-test was conducted to examine any difference in the mean score of the hesitancy to get the vaccine for the intervention group and control group before and after the program.”

13. This paragraph should be transferred from the Results section to the Methods section: “Concerning COVID-19 vaccination, related samples McNemar test was conducted to examine any vaccine rate difference before and after the program.”

14. The last paragraph in the Results section should be transferred to the Discussion and Conclusion sections of the manuscript: “To conclude, there was no significant difference in the hesitancy of COVID-19 vaccination …………………… of COVID-19 vaccination and increasing the tendency to get the vaccine in the interventional group only.”

Discussion:

15. The discussion should be extended with more details to include more information about vaccination hesitancy and determinants in developing countries, particularly among women planning to be pregnant, and how this study will solve this significant issue. Also, as previously mentioned, I prefer to include the benefits of vaccination over traditional precautionary measures, such as curfews, isolation, sealing the borders …..etc. In addition, this study’s findings should be compared with previous studies in the literature.

I would refer you to the following helpful references:

a. Al-Mistarehi AH, Kheirallah KA, Yassin A, et al. Determinants of the willingness of the general population to get vaccinated against COVID-19 in a developing country. Clin Exp Vaccine Res. 2021;10(2):171-182. doi:10.7774/cevr.2021.10.2.171

b. Wainstock T, Sergienko R, Orenshtein S, Sheiner E. Factors associated with COVID-19 vaccination likelihood during pregnancy [published online ahead of print, 2023 Jan 18]. Int J Gynaecol Obstet. 2023;10.1002/ijgo.14680. doi:10.1002/ijgo.14680

c. Blakeway H, Prasad S, Kalafat E, et al. COVID-19 vaccination during pregnancy: coverage and safety. Am J Obstet Gynecol. 2022;226(2):236.e1-236.e14. doi:10.1016/j.ajog.2021.08.007

d. Bechini A, Moscadelli A, Pieralli F, et al. Impact assessment of an education course on vaccinations in a population of pregnant women: a pilot study. J Prev Med Hyg. 2019;60(1):E5-E11. Published 2019 Mar 29. doi:10.15167/2421-4248/jpmh2019.60.1.1093

16. Study strengths:

a) I do not think the study sample size is one of the strengths.

b) One of its strengths is the study’s quasi-experimental interventional design, timeliness, and target population of women who did not take the COVID-19 vaccine.

c) Most vitally, this is one of the first studies that obtained insights into the effect of a tele-educational program on vaccine hesitancy among pregnant, planning for pregnancy, or breast-feeding mothers and examined detailed COVID-19 vaccine acceptance among such a vulnerable group in an Eastern Mediterranean developing country. Thus, this study tried to fill the gap in the literature regarding such issues outside Western countries.

17. Study limitations:

A. The study sample size is relatively small. However, the sample was collected from different areas throughout the country.

B. The selection bias can not be excluded as the study participants were collected from prenatal/maternity clinics. Thus, the results are unlikely to be generalizable beyond those who responded.

C. Also, the point mentioned in the manuscript: “Although the study aimed to include only pregnant women, planning for pregnancy, and breast-feeding mothers, their answers may vary from the general population and people in other situations.”

7. PLOS authors have the option to publish the peer review history of their article (what does this mean?). If published, this will include your full peer review and any attached files.

Reviewer #1: No

Reviewer #2: **Yes: **Abdel-Hameed Al-Mistarehi

---

## [Author Response · Author response to Decision Letter 1]

15 Feb 2023

Responses are provided in a separate file

---

## [Editor Report · Decision Letter 2]

20 Feb 2023

The Effect of COVID-19 Vaccine Tele-educational Program on Vaccine Hesitancy and Receiving the Vaccine Among Women Planning for Pregnancy, Pregnant or Breast-feeding Mothers

PONE-D-22-31498R2

Dear Dr. Masa'Deh,

We’re pleased to inform you that your manuscript has been judged scientifically suitable for publication and will be formally accepted for publication once it meets all outstanding technical requirements.

Kind regards,

Mohamad Abdullah, M.D., CHM.G/S., MRCS.

Academic Editor

PLOS ONE
---

## [Editor Report · Acceptance letter]

27 Feb 2023

PONE-D-22-31498R2 

The Effect of COVID-19 Vaccine Tele-educational Program on Vaccine Hesitancy and Receiving the Vaccine Among Women Planning for Pregnancy, Pregnant or Breast-feeding Mothers 

Dear Dr. Masa'Deh:

I'm pleased to inform you that your manuscript has been deemed suitable for publication in PLOS ONE. Congratulations! Your manuscript is now with our production department. 

Kind regards, 

on behalf of

Dr. Mohamad Abdullah 

Academic Editor

PLOS ONE